# Octahedral Molybdenum Cluster-Based Nanomaterials for Potential Photodynamic Therapy

**DOI:** 10.3390/nano12193350

**Published:** 2022-09-26

**Authors:** Marina Rodrigues Tavares, Kaplan Kirakci, Nikolay Kotov, Michal Pechar, Kamil Lang, Robert Pola, Tomáš Etrych

**Affiliations:** 1Institute of Macromolecular Chemistry of the Czech Academy of Sciences, Heyrovského Náměstí 2, 162 06 Prague 6, Czech Republic; 2Institute of Inorganic Chemistry of the Czech Academy of Sciences, 250 68 Husinec-Řež 1001, Czech Republic

**Keywords:** polymer carrier, photodynamic therapy, octahedral molybdenum clusters

## Abstract

Photo/radiosensitizers, such as octahedral molybdenum clusters (Mo_6_), have been intensively studied for photodynamic applications to treat various diseases. However, their delivery to the desired target can be hampered by its limited solubility, low stability in physiological conditions, and inappropriate biodistribution, thus limiting the therapeutic effect and increasing the side effects of the therapy. To overcome such obstacles and to prepare photofunctional nanomaterials, we employed biocompatible and water-soluble copolymers based on *N*-(2-hydroxypropyl)methacrylamide (pHPMA) as carriers of Mo_6_ clusters. Several strategies based on electrostatic, hydrophobic, or covalent interactions were employed for the formation of polymer-cluster constructs. Importantly, the luminescent properties of the Mo_6_ clusters were preserved upon association with the polymers: all polymer-cluster constructs exhibited an effective quenching of their excited states, suggesting a production of singlet oxygen (O_2_(^1^Δ_g_)) species which is a major factor for a successful photodynamic treatment. Even though the colloidal stability of all polymer-cluster constructs was satisfactory in deionized water, the complexes prepared by electrostatic and hydrophobic interactions underwent severe aggregation in phosphate buffer saline (PBS) accompanied by the disruption of the cohesive forces between the cluster and polymer molecules. On the contrary, the conjugates prepared by covalent interactions notably displayed colloidal stability in PBS in addition to high luminescence quantum yields, suggesting that pHPMA is a suitable nanocarrier for molybdenum cluster-based photosensitizers intended for photodynamic applications.

## 1. Introduction

Photodynamic therapy (PDT) represents a very promising therapeutic modality which employs the light irradiation as external stimuli to activate compounds, so-called photosensitizers (PS), for the treatment of various malignant tumors. The crucial mechanism of this therapy is based on the interaction of a light-activated PS with molecular oxygen dissolved in the intracellular medium, thus producing reactive oxygen species, e.g., singlet oxygen, O_2_(^1^Δ_g_), which can damage critical cellular components. Their cytotoxicity affects cell’s DNA, RNA, lipids, and proteins, resulting in tumor cell destruction. Selection of an appropriate PS is one of the most important factors influencing the efficacy of PDT. Various classes of compounds were described as PS for PDT, e.g., porphyrins, phtalocyanines, indocyanines, or bodipy dyes [1,2,3]. Nevertheless, the pharmacokinetics of PS plays a key role in the PDT treatment efficacy. Indeed, several research groups reported the encapsulation of numerous PS compounds to supramolecular systems, including polymer nanoparticles [4,5] and polymer micelles [4,6], and their controlled delivery. In particular, PS can be physically entrapped into nanoparticles by hydrophobic or electrostatic interactions of PS with biodegradable polymers, namely, poly(glycolic acid) [4,7], poly(lactic acid) [4,7], poly(lactic-*co*-glycolic acid) [8] and poly(ethylene glycol) [9].

Recently, octahedral molybdenum cluster compounds (Mo_6_) were reported as relevant photo/radiosensitizers for PDT [10,11,12,13,14], X-ray induced PDT [14,15,16,17,18], and photoinactivation of bacteria [11,19,20,21,22]. Mo_6_ clusters are nanometer-sized metallic aggregates where the distances between the atoms are similar to those found in corresponding bulk metals, evidencing electron delocalization on the whole cluster. For stabilization, the clusters are surrounded by eight strongly bonded inner ligands (L^i^), generally halogen (Cl, Br, I), and six apical ligands (L^a^) that can be of either organic or inorganic nature to form a cluster denoted [Mo_6_L^i^_8_L^a^_6_]^n^. Upon excitation with UV, blue light, or X-rays, these complexes form long-lived triplet states that relax via a broad red-NIR luminescence. This emission is efficiently quenched by oxygen leading to the formation of O_2_(^1^Δ_g_) in high yields [23,24]. Even though a previous study showed the phototoxic activity of Mo_6_ complexes against the human cervical cancer cell line HeLa [25], their limited solubility and/or low stability in physiological conditions, as well as their lack of tumor selectivity, may result in a systemic toxicity, thus restricting their possible use in PDT [26].

To overcome these shortcomings, we hypothesized that the interaction of Mo_6_ with a suitable hydrophilic and biocompatible polymer carrier based on *N*-(2-hydroxypropyl)methacrylamide (pHPMA) may improve both their stability in physiological conditions and also pharmacokinetic properties. The polymer conjugation may prolong half-life in circulation, significantly decrease interaction with serum proteins, promote a superior tumor specific accumulation, and reduce possible adverse effects, thus opening possibilities for their real clinical application. Previous studies have shown that PS molecules bound to pHPMA copolymers accumulate in the tumor via the enhanced permeability and retention (EPR) effect [1], in which the principle relies on the tumor tissues defective blood vessels and leaky vasculature combined with the poor lymphatic drainage, ensuring the O_2_(^1^Δ_g_) production in the tumor rather than in healthy tissues [2,27].

In this work, we describe the design, synthesis, and thorough evaluation of physico-chemical properties of HPMA-based polymer-Mo_6_ constructs. We aimed to explore various strategies, such as covalent or non-covalent interactions, to bind Mo_6_ clusters to the polymer carriers and to optimize their structures for possible future applications in the field of polymer therapeutics, more specifically on tumor-targeted photodynamic therapy. In this study, the term “complexes” is employed for the non-covalent constructs based on electrostatic and hydrophobic interactions, while the term “conjugates” is used for the constructs prepared by the covalent azide-alkyne “click chemistry” (Figure 1). The relationship between the selected coupling strategy and properties of polymer-Mo_6_ constructs is studied and described.

## 2. Materials and Methods

### 2.1. Materials

1,4-Dioxane, 2,2′-azo*bis*isobutyronitrile (AIBN), 2-cyanopropan-2-yl dithiobenzoate (CTA-AIBN), 2-thiazoline-2-thiol, 2,4,6-trinitrobenzene-1-sulfonic acid (TNBSA), 3-azido-1-propylamine, 4,4′-azo*bis*(4-cyanopentanoic acid) (ACVA), 4-cyano-4-(thiobenzoylthio)pentanoic acid (CTA-ACVA), dimethyl sulfoxide (DMSO), methacryloyl chloride, *N*,*N*-diisopropylethylamine (DIPEA), *N*,*N*-dimethylacetamide (DMA), phosphate buffered saline (phosphate buffer 0.01M and NaCl 0.154M, pH 7.4) (PBS), and *t*-butanol were purchased from Sigma-Aldrich (Prague, Czech Republic). 2,2′-Azo*bis*(4-methoxy-2,4-dimethylvaleronitrile) (V-70) was from Fujifilm Wako Chemicals Europe (Neuss, Germany). 3-Amino-1-(11,12-didehydrodibenzo[b,f]azocin-5(6H)-yl)propan-1-one (DBCO-NH_2_) was from Click Chemistry Tools (Scottsdale, AZ, USA). *N*-(3-*tert*-butoxycarbonyl-aminopropyl)methacrylamide (APMA-Boc) was purchased from Polysciences, Inc. (Warrington, PA, USA), and 1-aminopropan-2-ol was from TCI Europe (Zwijndrecht, Belgium). All solvents and chemicals were of analytical grade.

### 2.2. Synthesis of Monomers 

The monomers HPMA [28], 3-methacrylamidopropanoylthiazolidine-2-thione (Ma-AP-TT) [29] and cholest-5en-3*β*-yl 6-methacrylamido hexanoate (MA-AH-cholesterol) [30] were prepared according to the literature. High Performance Liquid Chromatography (HPLC) analysis was employed to verify the purity of monomers and chain transfer agents. Analysis was performed using a Shimadzu HPLC system with a C18 reversed-phase Chromolith Performance RP-18e column and a diode array detector (Shimadzu SPD-M20A), using water/acetonitrile (gradient of 5–95% *v*/*v* acetonitrile) as eluent with 5 mL min^−1^ flow rate.

### 2.3. Synthesis of Polymer Precursors P1–P5

Statistical copolymers were prepared by reversible addition–fragmentation chain transfer (RAFT) polymerization of HPMA with respective monomers as follows: MA-AP-TT was employed for **P0a**, **P0b**, **P1**, and **P5** copolymers; APMA-Boc for **P2 [31]**; MA-AH-cholesterol for **P3**; MA-AH-cholesterol and APMA-Boc for **P4**. The chain transfer agent CTA-AIBN was used for all precursors except for **P2**, in which CTA-ACVA was employed instead. Reaction conditions were adapted from our previous studies [32], using a mixture of *t*-butanol and DMA for all precursors except for **P2**, in which a mixture of water and dioxane was employed. Dithiobenzoate (DTB) groups originating from CTA were removed by reaction with an excess of AIBN as previously described [33]. For **P2** and **P4**, *t*-butoxycarbonyl (Boc) groups were removed by heating in distilled water at 150 °C for 1 h [34]. Copolymer **P1** with COOH groups along the polymer chain was prepared via hydrolysis of thiazolidine-2-thione (TT) groups of the polymer precursor poly(HPMA-*co*-MA-AP-TT) in phosphate buffer (pH 8.0). Detailed structures of copolymer precursors **P1**–**P5** are shown in Figure 2, and their physico-chemical characterization is described in Table 1. Detailed synthetic procedures and the ratio between monomers, chain transfer agents, and initiators are described in Appendix A. ^1^H NMR spectra of polymer precursors **P3** and **P4** are shown in Appendix A.

### 2.4. Synthesis of Mo_6_ Clusters C1–C4

Previously published procedures were employed to prepare Mo_6_ cluster compounds: [Mo_6_I_8_(OCOC_4_H_8_PPh_3_)_6_]Br_4_ (**C1**) [12], Na_2_[Mo_6_I_8_(OPOPh_2_)_6_] (**C2**) [11], Na_2_[Mo_6_I_8_(cholate)_6_] (**C3**) [35], and Na_2_[Mo_6_I_8_(N_3_)_6_] (**C4**) [36]. Schematic structures of clusters **C1**–**C4** are shown in Figure 2. For characterization see Table 2.

### 2.5. Synthesis of Polymer-Cluster Constructs POL1–POL6

Different conjugation strategies were exploited aiming to optimize the constructs’ structure for possible future applications in the field of polymer therapeutics. In this study, the term “complexes” is employed for the non-covalent constructs based on electrostatic (**POL1–****POL2**) and hydrophobic (**POL3**–**POL4**) interactions, while the term “conjugates” is used for the constructs using covalent attachment of the Mo_6_ clusters to the polymer backbone (**POL5–POL6**). The physico-chemical characterization of polymer-cluster constructs **POL1**–**POL6** is shown in Table 3, and their detailed synthetic procedures are described below.

#### 2.5.1. Electrostatic Non-Covalent–**POL1** and **POL2**

Solutions of the polymer precursor and the cluster, each in 100 µL of methanol, were prepared separately. In case of cluster solution, addition of 10 µL of DMSO was necessary for complete dissolution. Aliquots of 50 µL from each solution were mixed together and vortexed for 10 min. The reaction mixture was added into 1 mL of distilled water and kept under stirring at room temperature for 1 h. Then, methanol was removed under vacuum, and water was added to adjust the volume to 1 mL. The resulting solutions were used for dynamic light scattering (DLS) and quantum yield (QY) measurements. **POL1** was composed of **P1** (10 mg, 1.3 µmol of -COOH) and cluster **C1** (1.3 mg, 0.32 µmol) while **POL2** was composed of **P2** (10 mg, 2.35 µmol of -NH_2_) and cluster **C2** (3.5 mg, 1.19 µmol). 

#### 2.5.2. Hydrophobic Non-Covalent Complexes–**POL3** and **POL4**

The procedure for preparation of **POL3** and **POL4** was analogous to the one described in 2.5.1., except that the addition of DMSO was not necessary for dissolution of the cluster **C3**. **POL3** was composed of **P3** (10 mg) and cluster **C3** (2.5 mg) while **POL4** was composed of **P4** (10 mg) and cluster **C3** (2.5 mg).

#### 2.5.3. Covalent Conjugates–**POL5**–**POL6**

A solution of **P5** (300 mg, 145 µmol of DBCO groups) in dry methanol (2.4 mL) was added into a solution of cluster **C4** (49.6 mg, 145 µmol of azide groups) in 0.75 mL of dry methanol under stirring. The reaction mixture was maintained under argon for 30 min; then, it was stirred at 25 °C overnight. Isolation and purification procedures were performed as follows: first, the polymer was precipitated into a mixture of acetone/diethyl ether (1/1) twice. The product was washed with pure diethyl ether and dried under vacuum, affording the polymer-cluster conjugate **POL5** (296 mg, 93%). HPLC analysis was performed to control the course of reactions. **POL6** was prepared analogously to **POL5**; however, part of DBCO groups was reacted with 3-azido-1-propylamine via a copper-free alkyne-azide cycloaddition as follows: 3-azido-1-propylamine (1.3 μL, 13.6 μmol) was added to a solution of **P5** (150 mg, 72.5 μmol of DBCO groups) in dry DMA (2 mL) followed by cluster **C4** (49.6 mg, 145 µmol of azide groups) in 0.75 mL of dry methanol. The reaction mixture was bubbled with argon; then, it was stirred for 3 h at 24 °C. The reaction mixture was diluted with methanol (2 mL) and purified using a Sephadex LH-20 column with methanol elution and UV detection. The conjugate-containing fraction was collected and concentrated under vacuum to 2 mL. The polymer was isolated by precipitation into the mixture of acetone and diethyl ether (2/1; 100 mL) and dried to yield **POL6** (136 mg; yield 91%).

### 2.6. Physico-Chemical and Photophysical Characterization

#### 2.6.1. Size Exclusion Chromatography (SEC)

The number-average molecular weight (*M*_n_), weight-average molecular weight (*M*_w_), and dispersity (*Ð*) of polymer precursors **P1**–**P5** and polymer-cluster conjugates **POL5** and **POL6** were determined by a Shimadzu HPLC system equipped with a size exclusion chromatography (SEC) column TSK 3000 SWXL column (Tosoh Bioscience, Tokyo, Japan). Evaluation was carried out using a multi-angle light scattering (MALS) DAWN HELEOS II (Wyatt Technology Co., Santa Barbara, CA, USA), photodiode array SPD-M20A (Shimadzu, Japan) and differential refractometer index Optilab^®^-rEX (Wyatt Technology Co., Santa Barbara, CA, USA) detectors. The analysis was performed using a mixture of methanol and 0.3 M sodium acetate buffer, pH 6.5 (4/1, *v/v*) as a mobile phase at a flow rate of 0.5 mL min^−1^. The ASTRA software (version 8.1, Wyatt Technology Co., Santa Barbara, CA, USA) was used for calculation of *M*_w_ and *Ð* values.

#### 2.6.2. Dynamic Light Scattering (DLS)

The hydrodynamic diameter (*D*_h_) and surface zeta potential (ZP) of all samples was measured using a Nano-ZS instrument (ZEN3600, Malvern, UK) with a laser wavelength of 632.8 nm, and the intensity of the scattered light was detected at an angle θ = 173°. Precursors were dissolved in PBS at 1 mg mL^−1^. Clusters and polymer-cluster constructs were dissolved in methanol, and aliquots of these solutions were added to deionized water (pH ~ 6) or PBS to obtain the final concentration of 1 mg mL^−1^; their long-term stability was evaluated. The values were determined as a mean of at least five independent measurements.

#### 2.6.3. UV–VIS Spectrophotometry

UV–VIS spectrophotometry (Specord 205 ST, Analytic Jena AG, Jena, Germany) was used for determination of the content of TT, DBCO, and amine groups. The molar absorption coefficient of *ε*(TT) = 10,300 L mol^−1^ cm^−1^ (λ_max_ = 305 nm) in methanol was used for determination of TT groups in the polymer precursors before hydrolysis (**P1**) or aminolysis (**P5**). In the case of DBCO groups, *ε*(DBCO) = 13,000 L mol^−1^ cm^−1^ (λ_max_ = 292 nm) in methanol was employed. The content of amine groups in polymer precursors **P2**, **P4**, and polymer-cluster conjugate **POL6** was determined using a modified TNBSA assay as published earlier [37]. For **P2** and **POL6**, the solution was prepared in borate buffer (0.1 M Na_2_B_4_O_7_·10H_2_O, pH 9.3) at the concentration of 2 mg mL^−1^. An aliquot of 100 μL of this solution was mixed with borate buffer (875 μL) and 0.03 M solution of TNBSA in water (25 μL). The molar absorption coefficient ε (NH_2_) = 17,200 L mol^−1^ cm^−1^ (λ_max_ = 500 nm) was used, and the absorbance was measured after 90 min of incubation. In the case of **P4**, a similar evaluation method was employed, but the sample was dissolved in a mixture of borate buffer and DMSO (9/1) due to the hydrophobic character of cholesterol moieties [30].

#### 2.6.4. Nuclear Magnetic Resonance (NMR) Spectroscopy

For determination of cholesterol content in polymer precursors **P3** and **P4**, ^1^H NMR spectra were measured with a Bruker Avance III 600 spectrometer (Bruker, Karlsruhe, Germany) operating at 600.2 MHz using DMSO-*d*_6_ as solvent. Typical conditions for measurements of the spectra were as follows: π/2 pulse width 10 µs, relaxation delay 10 s, spectral width 10 kHz, acquisition time 3.21 s, 200 scans, and 5 mm NMR tubes were used. The content of cholesterol moieties statistically distributed along the polymer backbone was assessed using the integral intensities of signals at *δ* ≈ 4.71 ppm (1 H, O*H*) and *δ* ≈ 3.67 ppm (1 H, C*H*) from the HPMA monomer unit and the integral intensity of the signal at *δ* ≈ 5.34 ppm (1 H, C*H*) from C-6 of cholesterol moiety. NMR data for both polymer precursors are shown in the Appendix A.

#### 2.6.5. Attenuated Total Reflectance (ATR) Fourier-Transform Infrared (FTIR) Spectroscopy

ATR FTIR spectra were recorded using a Thermo Nicolet Nexus 870 FTIR spectrometer (Bruker, Karlsruhe, Germany) purged with dry air and equipped with a liquid-nitrogen-cooled MCT (mercury cadmium telluride) detector. All the spectra were acquired using a Golden Gate single reflection ATR accessory (Specac Ltd., Orpington, UK) equipped with a diamond internal reflection element. Measurements were performed at room temperature using the following parameters: resolution 4 cm^−1^ and 256 scans. All data were processed in the OMNIC software (ver. 8.3.103). The atmosphere spectrum was subtracted from the acquired spectra, and then, the resulting spectra were subjected to the baseline and ATR corrections. Measurements were performed for solutions of **C4**, **POL5**, and **POL6** in distilled water at 0.2 wt. % of Mo_6_-cluster equivalent.

#### 2.6.6. Luminescence Spectroscopy

Absolute photoluminescence quantum yields and emission spectra in deionized water or PBS were measured using a Quantaurus QY C11347-1 spectrometer (Hamamatsu, Japan). The samples were prepared by adding small aliquots of concentrated methanol solutions of the polymer-cluster constructs to deionized water or PBS to reach the final concentration of 0.1 mg mL^−1^. All samples were excited at 400 nm. In order to perform measurements under various concentrations of dissolved oxygen, the aqueous solutions of the clusters and the corresponding polymer constructs were saturated with air or argon.

## 3. Results and Discussion

In order to prevent Mo_6_ clusters’ aggregation in aqueous solutions, we present novel synthetic strategies for the synthesis of biocompatible polymer-coated Mo_6_ cluster constructs by employing covalent or non-covalent interactions with various pHPMA copolymers differing in their structure. This study is focused on the physico-chemical and photophysical properties of the polymer-cluster constructs intended as nanomedicines for anti-tumor therapy.

### 3.1. Synthesis of Polymer Precursors

Controlled RAFT polymerization technique was employed in order to prepare well-defined polymer precursors with an appropriate number of functional groups, such as TT, COOH, NH_2_, and DBCO moieties, which are used for further chemical modifications. 

First, poly(HPMA-*co*-MA-AP-TT) (**P0a**) containing reactive TT groups were prepared and subsequently hydrolyzed in phosphate buffer (pH 8.0) to yield poly(HPMA-*co*-MA-AP-COOH) (**P1**). Even though much higher amounts of negatively charged groups may be necessary for stronger electrostatic interactions between the polymer and molybdenum molecule, highly, negatively charged systems tend to be captured by the reticuloendothelial system (RES), mainly in the liver and spleen, thus impairing their use for in vivo applications [38]. Therefore, a lower ratio of MA-AP-TT related to HPMA was used for the polymerization of **P0a** precursor, resulting in copolymers with 1.9 mol. % of final reactive groups and *M*_w_ ≈ 18,700 g·mol^−1^.

Another poly(HPMA-*co*-MA-AP-TT) (**P0b**) was prepared, but using a higher ratio of MA-AP-TT related to HPMA, as well as a higher amount of monomers related to CTA and initiator in the reaction mixture, resulting in a precursor containing 10.2 mol. % of reactive TT groups and *M*_w_ ≈ 39,200 g mol^−1^. Here, the respective content of functional groups was chosen aiming to afford a precursor with a higher number of moieties available for further covalent interactions. To prepare **P5** containing 8 mol. % of DBCO groups, **P0b** was reacted with an amine-functionalized DBCO. The copper-free click chemistry approach was selected to avoid the use of copper in further reactions, thus reducing toxicity-related risks and also avoiding time-consuming and complicated purification steps [39].

Higher amounts of positively charged groups on polymer precursors may be necessary to achieve stronger electrostatic interactions with negatively charged molybdenum clusters. Unfortunately, a higher content of amine groups is known to generate toxicity in vitro and in vivo; therefore, the ratio of comonomers was optimized to afford poly(HPMA-*co*-APMA-Boc) (**P2**) containing approximately 5.5 mol. % of amine groups.

The introduction of hydrophobic moieties, such as cholesterol or its derivatives, into the structure of the water-soluble polymer carrier switches the character of the polymers to their amphiphilic nature [30,40,41,42]. Such amphiphilic polymers can either self-assemble into the core-shell micellar structures or can form a coating of hydrophobic nanoparticles or liposomes via interaction of cholesterol moiety with hydrophobic compartments of those nanomaterials [43,44]. According to the described procedures [30,40], the amphiphilic polymer precursors **P3** and **P4** were prepared containing 2.3 and 2.5 mol. % of cholesterol moieties, respectively, and comparable molecular weights around 25,000 g mol^−1^ with narrow dispersity. For ^1^H-NMR spectra of **P3** and **P4**, see Appendix A, respectively. The introduction of certain number of amine groups into the polymer precursor **P4** was performed to adjust the negative charge of C3, thus improving the properties of the system for eventual in vivo applications. According to the published data, neutral nanomedicines exhibit more favorable pharmacokinetic properties such as plasma half-life, recognition by RES, and possible adhesion to vascular endothelium [38].

Linear precursors **P1** and **P2** exhibited hydrodynamic diameters in aqueous solution around 5 nm, typical for HPMA-based polymer random coils, whereas the diameter of **P5** was significantly higher (*D*_H_ ≈ 11 nm) indicating eventual association of the macromolecules due to the presence of the hydrophobic pendant DBCO group and a higher molecular weight of the polymers. As expected, **P3** showed the highest hydrodynamic diameter (*D*_H_ ≈ 40 nm) due to the hydrophobic characteristic of pendant cholesterol moieties, which enable **P3** to form micelles in aqueous solution, as already observed in our previous studies of such HPMA copolymers with cholesterol derivatives [40,41]. Nevertheless, positively charged amine moieties distributed along the same polymer backbone (**P4**) may have contributed to the formation of smaller micelles (*D*_H_ ≈ 27 nm), probably due to the increased hydrophilicity of the precursor, hence impairing the formation of larger micelles.

Detailed structures and physico-chemical characterization of the polymer precursors **P1**–**P5** are shown in Figure 2 and Table 1, respectively.

### 3.2. Preparation of Polymer-Cluster Constructs

In this study, several conjugation strategies were exploited aiming to optimize the constructs’ structure for possible future applications in the field of polymer therapeutics. Our initial attempts relied on the use of non-covalent electrostatic and hydrophobic interactions for preparation of the non-covalent constructs (**POL1**–**POL2**) and (**POL3**–**POL4**), respectively, which are referred to as “complexes”. Afterwards, the covalent attachment of the Mo_6_ clusters was employed to prepare **POL5**–**POL6**, which are called by the term “conjugates”.

The structures of precursors **P1**–**P4** and clusters **C1–C3,** used for non-covalent complex formation, and precursor **P5** and cluster **C4,** used for preparation of the covalent conjugates, are summarized in Figure 2.

Polymer-cluster complexes **POL1** and **POL2** were prepared by employing electrostatic interactions between **P1** and **P2** with the positively and negatively charged Mo_6_ clusters **C1** and **C2**, respectively. An equimolar ratio between the oppositely charged functional groups of the polymer precursors and clusters was used. Consequently, this resulted in various contents of the clusters within their respective complexes: **POL1** (11.5 wt%, cluster charge 4+) and **POL2** (25.9 wt%, cluster charge 2−). **POL1** exhibited ZP value +9 mV which was slightly lower than that of cluster **C1** (+13 mV) as a consequence of the electrostatic complex formation with the negatively charged polymer precursor **P1**. Although the change in ZP value was relatively low, the formation of the complex **POL1** was accompanied with a dramatic change of the hydrodynamic diameter from 48 nm of the cluster **C1** (that tends to aggregate in water) to 5 nm of the polymer-coated complex **POL1**. The cluster–polymer interaction in this particular case enabled to obtain the unimolecular complexes presented in single cluster molecules coated by the hydrophilic copolymer. Most probably, the cluster containing +4 charge is coated with one or two polymer chains affording a slightly positive complex and a small dimension in contrast to huge aggregates of **C1** cluster molecules. When the polymer precursor **P2** with amine groups was employed for the complex formation, a significant increase in the zeta potential of complex **POL2** (+4 mV), reaching almost neutral complex, was observed in comparison to the highly negatively charged cluster **C2** (−67 mV) used in complexation. Such behavior indicate that the cluster was successfully modified with the positively charged polymer. Importantly, the hydrodynamic size of complex **POL2** was significantly higher (*D*_h_ ≈ 30 nm) compared with **POL1** (*D*_h_ ≈ 5 nm). We hypothesize that this was caused by the presence of much higher content of amino groups on polymer than the carboxyl groups on the cluster. The polymer coated sufficiently the clusters, but one polymer chain was generally involved in the coating of more than one cluster molecule in solution, thus the crosslinking of the polymer-cluster **C2** occurred and increased the size.

Another non-covalent method of the polymer-cluster complex preparation explored in this study was based on hydrophobic interactions of polymer precursors **P3** and **P4**, both bearing hydrophobic cholesterol moieties, with the cholate-based cluster **C3**, affording the polymer-cluster complexes **POL3** and **POL4**, respectively. Loading of 20 wt. % of Mo_6_ cluster was chosen for both complexes **POL3** and **POL4.** The introduction of amine groups to polymer **P4** was performed to verify the influence of the positive charge upon the formation of the hydrophobic complexes (**POL3** and **POL4**). After interaction of precursor **P3,** bearing cholesterol moieties and neutral charge, with **C3** (ZP = −9 mV), the zeta potential of the resulting complex **POL3** slightly dropped to −14 mV. Opposite to that, a neutral to slightly positive surface charge (+1 mV) was observed when amine groups were introduced into the structure of **POL4**, showing the benefit caused by the addition of amino groups to the complex. The hydrodynamic sizes of original **P3** and **P4** polymers, *D*_h_ ≈ 40 and 27 nm, respectively, showed formation of the micelles self-assembled from these amphiphilic copolymers. Importantly, after the complexation with **C3**, both complexes **POL3** and **POL4** showed smaller hydrodynamic sizes, *D*_h =_ 8.4 and 12 nm, respectively, in comparison to their polymer precursors. We summarize that the observed change in size upon the addition of the hydrophobic moieties containing cluster is most probably caused by disruption of the self-assembled micellar vesicles formed by the amphiphilic polymer precursors and subsequent rearrangement of the polymer chains around the hydrophobic cluster in aqueous solution. In contrast to **POL2** and similarly to **POL1**, both **POL3** and **POL4** are rather formed by a single molecule of cluster coated by a small number of polymer chains. Physico-chemical characteristics of Mo_6_ clusters are shown in Table 2.

With respect to the limited colloidal stability of the non-covalent polymer-cluster complexes, which will be further discussed within this manuscript, we evaluated the covalent binding of Mo_6_ clusters to the polymer carriers as an alternative approach. For this purpose, DBCO groups from polymer precursor **P5** were used for the attachment of Mo_6_ cluster **C4** via a copper-free alkyne-azide cycloaddition (“click reaction”), affording stable covalent polymer-cluster conjugates **POL5** and **POL6**. For both conjugates, the content of Mo_6_ cluster was calculated considering the ratio 1/1 between DBCO groups on polymer precursor and azide groups on cluster moiety.

It is generally known that the highly positively charged systems should be toxic for the body cells and can adhere to the negative vascular endothelium, leading to a lower concentration in the plasma along with an impaired EPR effect [38]. On the other hand, highly negatively charged systems are easily recognized taken up by the immune system, and then stacked in the kidneys. To avoid such effects and potential drawbacks, the ideal surface charge should be neutral or only slightly negative. Therefore, additional amine groups were introduced into the polymer-cluster conjugate **POL6** via reaction of the conjugate **POL5** with 3-azido-1-propylamine to neutralize the negative charge on the cluster and make the conjugate neutral in charge. 

Infrared spectroscopy was used to measure the conversion of the alkyne-azide cycloaddition. The efficiency of **C4** conjugation to **P5** was determined using the strong asymmetric stretching vibration mode of azide groups [45,46,47]. To assess the amount of unreacted azide groups, a reference was set up as the integral area of the asymmetric stretching vibrational band of azide groups centered at 2056 cm^–1^ of **C4** (0.2 wt% Mo_6_-cluster in distilled water; Figure 3, dark grey line) [45,46]. This typical complex profile can be attributed to weak Fermi resonance of the vibration with combination tones of C–N stretching vibrations and partly to possible differences in local environments of the six azide groups [47,48]. The azide band’s shift to a lower wavenumber (2056 cm^–1^) compared to the typical position described in the literature was attributed to the presence of charged molybdenum and iodine atoms in the cluster’s structure [47]. Additionally, the cluster forms a stable colloid in water, and azide groups are involved in H-bonding with water molecules [48] which may contribute to the observed red-shifted position of the band.

After conjugation of **C4** with the precursor **P5**, affording **POL5**, and further introduction of amine groups along the polymer backbone affording **POL6,** the asymmetric stretching band intensity of the azide groups greatly decreased (Figure 3, blue and red lines) due to newly formed covalent bond. As the bands’ position remained in the same region, the peak area tool in the OMNIC software was used to compare their intensities in the spectra of **POL5** and **POL6** using **C4** band as a reference. After the conjugation reaction, 9% ± 1% of azide groups remained unreacted for **POL5,** and similar value was found for **POL6**. This assessment indicates that the click reaction was successful as majority of the azide groups reacted.

The zeta potential of **POL5** conjugate (−17 mV) was similar to the respective cluster **C4** (−16 mV). As mentioned above, negatively charged nanoparticles could exhibit unfavorable pharmacokinetic properties after the body injection. Therefore, to avoid such behavior, part of the DBCO groups of polymer precursor **P5** was reacted with 3-azido-1-propylamine prior to addition of cluster **C4** to compensate, at least partially, the negative zeta potential of the polymer-cluster conjugate. Such combinational approach provided the polymer precursor with 1.5 mol. % of amine groups along the polymer backbone and consequently polymer-cluster conjugate **POL6** with zeta potential adjusted to −7 mV, which is much more suitable for future therapeutic application of the developed polymer-cluster conjugates. Both *M*_w_ and *D*_h_ values proved the formation of polymer-cluster conjugates. The *M*_w_ slightly increased for **POL5** and **POL6** (*M*_w_ ≈ 49,500 g mol^−1^ and 52,000 g mol^−1^, respectively) when compared to their precursor **P5** (*M*_w_ ≈ 40,000 g mol^−1^) upon introduction of Mo_6_ clusters in the structure. The hydrodynamic size remains similar to that of the other complexes, thus most probably forming the unimolecular complexes presented with a single or low number of cluster molecules. For physico-chemical and photophysical characteristics of polymer-cluster constructs **POL1**–**POL6**, see Table 3.

Importantly, we can summarize that we have successfully designed and synthesized several polymer-cluster systems differing in their inner structure and mode of the polymer-cluster interaction or bonding.

### 3.3. Stability and Photophysical Properties of the Polymer-Cluster Constructs

We further evaluated the colloidal stability and photophysical properties of the polymer-cluster constructs in PBS, a biologically relevant medium. In the case of **POL1** and **POL2**, formed by electrostatic interactions, the complexes were not stable in any manner as the electrostatic interactions were not strong enough to keep the polymer-cluster complexes stable. **POL1** and **POL2** complexes in PBS immediately formed huge aggregates, and their precipitation was observed quite rapidly after their dissolution. In the case of **POL3** and **POL4**, exploiting the hydrophobic interactions, a slightly better stability in PBS was observed. Nevertheless, aggregates were also formed, suggesting that the hydrophobic polymer-cluster interactions were disrupted in PBS. Consequently, it was not possible to properly perform DLS, SEC or luminescence spectroscopy studies in PBS for any of these non-covalent complexes.

In terms of photophysical properties, the constructs were first studied in deionized water, where they displayed the typical broad emission band of the Mo_6_ clusters with maxima in the 685–697 nm range (Figure 4). The emissivity was high with luminescence quantum yield ranging from 0.16 for **POL1** to 0.49 for **POL4** in argon saturated water (see Table 3). The quenching of the emission by oxygen was efficient with a fraction of triplet states quenched by oxygen in an air saturated solution (*F_T_* = 1 − *Φ_L_*(air)/*Φ_L_*(Ar)) of approximately 0.8 for all constructs, indicating good accessibility of the clusters to dissolved oxygen. This feature suggests an effective production of O_2_(^1^Δ_g_) which is a major factor for a successful photodynamic treatment. Overall, the luminescent properties of the clusters were preserved upon association with their respective polymers.

In contrast to the non-covalent complexes which were not stable in PBS buffer, covalent polymer-cluster conjugates **POL5** and **POL6** displayed remarkable stability in PBS. Their hydrodynamic diameter of approximately 7–11 nm and ZP values did not change significantly even after 5 days (see Table 4). Additionally, no drastic changes were observed in the photophysical properties of the conjugates dissolved in PBS in comparison with deionized water solutions, except for slightly red shifted emission maxima and higher quantum yields (Table 4). The photophysical stability of the solutions was evaluated over a five-day period revealing no significant changes in the emission maxima, quantum yields, and oxygen quenching constant, which evidenced the high stability of the photosensitizing system (see Figure 5) in PBS. These features are attractive for photodynamic applications as a reasonable stability of the photosensitizing system is required for an effective PDT. 

Taken all together, even though suitable photophysical properties were found for the polymer-cluster complexes, their colloidal stability in PBS does not allow further in vitro and in vivo testing and application. For polymer-cluster conjugates **POL5** and **POL6** bearing the Mo_6_ cluster covalently bound to the polymer carrier, a very good stability and photophysical properties were observed. Both size and zeta potential were maintained after several days in PBS and even after months of storage as dried powders; hence, we believe that these conjugates are more suitable for further biomedical applications.

## 4. Conclusions

This study described the design and synthesis of hydrophilic HPMA-based polymer constructs with Mo_6_ clusters, potent singlet oxygen photosensitizers, and their structures’ optimization from the chemical and physico-chemical point of view. We investigated three methods of preparation of the constructs using electrostatic, hydrophobic, or covalent interactions between the polymer backbone and cluster moieties. The luminescent properties of the Mo_6_ clusters were preserved upon association with their respective polymers and all polymer-cluster constructs exhibited a production of O_2_(^1^Δ_g_), which is a major factor for a successful photodynamic treatment. The conjugates prepared by covalent interactions, such as the azide-alkyne “click chemistry”, were the best in the series–they possessed a high colloidal stability in PBS and provided high luminescence quantum yields. Moreover, a significant advantage of the synthesis is the fact that copper is completely avoided during the procedure. Results from physico-chemical and photophysical evaluation indicate that the conjugates with Mo_6_ covalently attached to the polymer backbone are prospective candidates for biological evaluation including measurements of the cytotoxicity effect against selected cancer cells and in vivo experiments using suitable animal tumor models to assess the PDT efficacy of such system.

## Figures and Tables

**Figure 1 nanomaterials-12-03350-f001:**
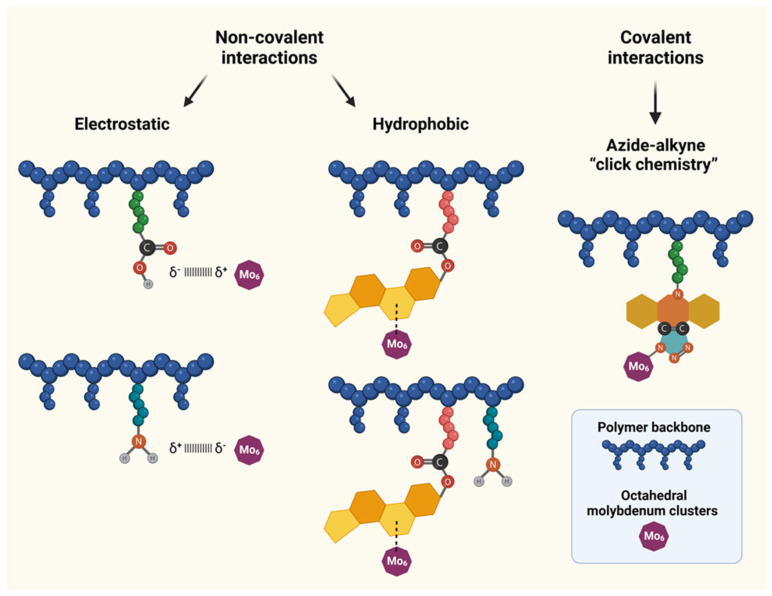
Schematic representation of non-covalent and covalent interaction approaches employed for binding of Mo_6_ clusters to polymer precursors; Mo_6_ cluster not-to-scale.

**Figure 2 nanomaterials-12-03350-f002:**
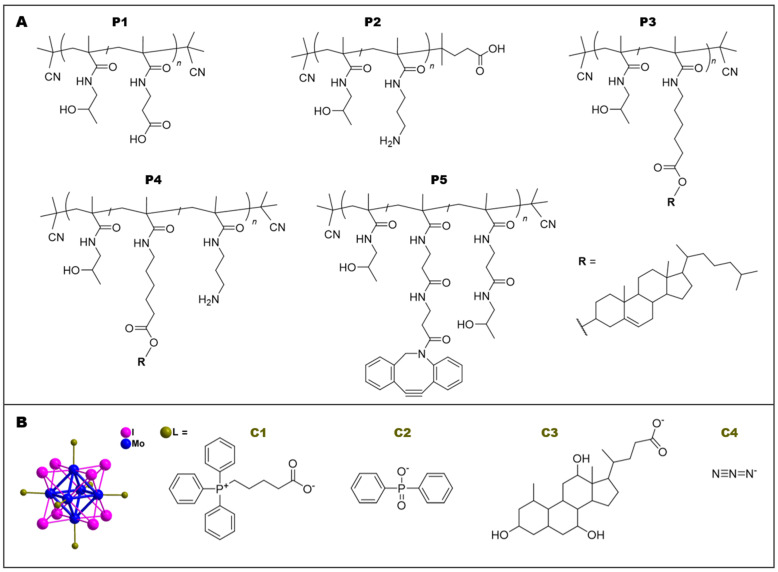
Polymer precursors and Mo_6_ clusters used for non-covalent and covalent interactions: (**A**) chemical structures of precursors **P1**–**P5**; (**B**) schematic representation of the molecular structure of [M_6_L^i^_8_L^a^_6_]^n^ cluster and ligands for clusters **C1–C4**; color coding: molybdenum (blue), iodine (L^i^: magenta), apical ligands (L^a^: green); hydrogen atoms are omitted for clarity.

**Figure 3 nanomaterials-12-03350-f003:**
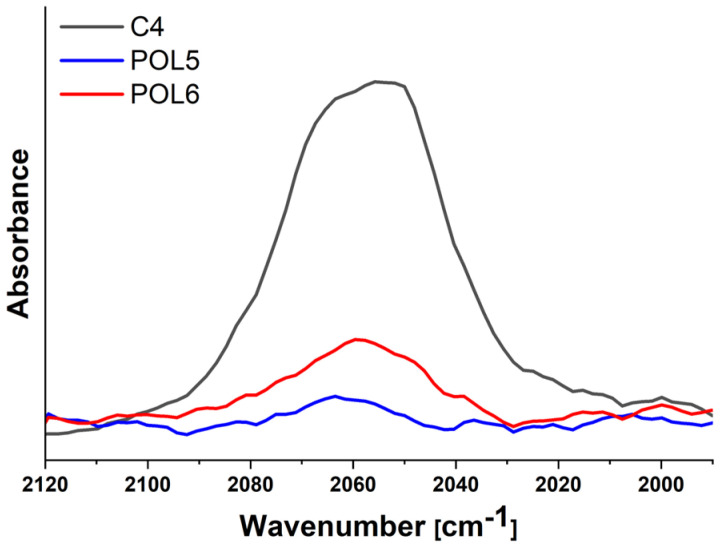
The region of stretching vibrations of the azide groups of cluster **C4** and polymer-cluster conjugates **POL5** and **POL6** in ATR FTIR spectra after subtraction of the corresponding spectrum of water.

**Figure 4 nanomaterials-12-03350-f004:**
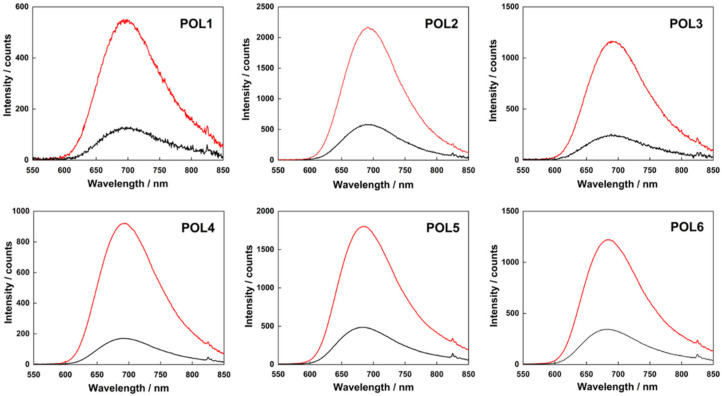
Luminescence emission spectra of **POL1**–**POL6** in deionized water: argon-saturated (red curve) and air-saturated (black curve) dispersions. All samples were excited at 400 nm.

**Figure 5 nanomaterials-12-03350-f005:**
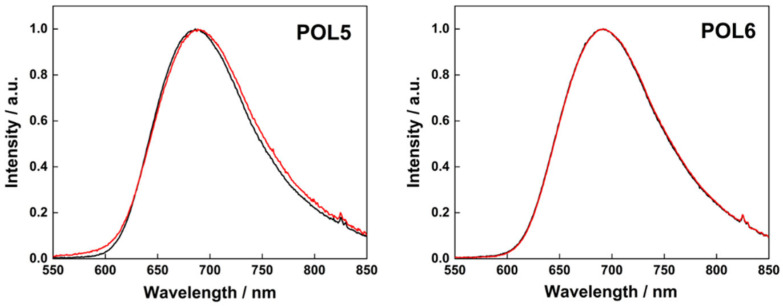
Normalized luminescence emission spectra of **POL5** and **POL6** in argon-saturated PBS. Solutions were fresh (black curve) or 5 days old (red curve). All samples were excited at 400 nm.

**Table 1 nanomaterials-12-03350-t001:** Physico-chemical characteristics of the polymer precursors.

Polymer Precursor	Structure	Functional Group	Functional Groups (mol. %) *^a^*	*M*_w_ (g/mol) *^b^*	*Ð ^b^*	*D*_H_ (nm) *^c^*
**P0a** **P0b**	poly(HPMA-*co*- MA-AP-TT)	TT	(a) 1.9 (b) 10.2	18,500 39,200	1.03 1.04	5.9 ± 0.7
						7.4 ± 0.4
**P1**	poly(HPMA-*co*- MA-AP-COOH)	COOH	1.9	18,700	1.05	5.3 ± 0.2
**P2**	poly(HPMA-*co*-APMA)	NH_2_	5.1	24,100	1.04	4.9 ± 0.1
**P3**	poly(HPMA-*co*- MA-Acap-cholesterol)	cholesterol	2.3	26,400	1.06	39.9 ± 1.1
**P4**	poly(HPMA-*co*-APMA-*co*- MA-AH-cholesterol)	cholesterol + NH_2_	2.5 cholesterol 2.3 NH_2_	24,600	1.07	26.8 ± 0.7
**P5**	poly(HPMA-*co*- MA-AP-DBCO)	DBCO	8	40,000	1.06	11.5 ± 0.8

*^a^* Contents of TT, DBCO, and NH_2_ groups were evaluated by UV–VIS spectrophotometry; the content of cholesterol was determined using NMR Spectroscopy. *^b^* The weight-average molecular weight (*M*_w_) and dispersity (*Ð*) were determined using SEC with RI and MALS detection. The analysis was performed using a TSK 3000 SWXL column with methanol/0.3 M sodium acetate buffer, pH 6.5 (4/1, *v/v*), as a mobile phase. *^c^* The mean hydrodynamic diameter (*D*_H_) was obtained by DLS with intensity of scattered light detected at angle θ = 173° in PBS.

**Table 2 nanomaterials-12-03350-t002:** Physico-chemical characteristics of Mo_6_ clusters.

Cluster	Formula	*M*_w_ (g/mol)	*D*_H_ (nm) *^a^*	*ZP* (mV) *^a^*	Charge
**C1**	[Mo_6_I_8_(OCOC_4_H_8_PPh_3_)_6_]Br_4_	4084.9	48.4 ± 5.3	13	4
**C2**	Na_2_[Mo_6_I_8_(OPOCPh_2_)_6_]	2939.9	20.2 ± 11.6	−67	−2
**C3**	Na_2_[Mo_6_I_8_(cholate)_6_]	4082.0	5.9 ± 1.4	−9	−2
**C4**	Na_2_[Mo_6_I_8_(N_3_)_6_]	1889.0	60.6 ± 15.3	−16	−2

*^a^* The mean hydrodynamic diameter (*D*_H_) and zeta potential (*ZP*) were obtained by DLS measurement with intensity of scattered light detected at angle θ = 173° in deionized water (pH ~ 6).

**Table 3 nanomaterials-12-03350-t003:** Physico-chemical and photophysical characteristics of polymer-cluster constructs.

Polymer-Cluster Constructs	Prepared From	Type of Interaction	Mo Cluster (wt%)	*D*_H_ (nm) *^a^*	*ZP* (mV) *^a^*	*λ**_L_* (nm) *^b^*	*Φ_L_*(Ar) *^b^*	*Φ_L_*(air) *^b^*	*F_T_(air) ^b^*
**POL1**	P1 + C1	Electrostatic	11.5	5.1 ± 1.1	9	695	0.16	0.04	0.75
**POL2**	P2 + C2	Electrostatic	25.9	29.2 ± 9.0	4	690	0.39	0.08	0.79
**POL3**	P3 + C3	Hydrophobic	20.0	8.4 ± 2.3	−14	690	0.25	0.05	0.80
**POL4**	P4 + C3	Hydrophobic	20.0	12.0 ± 3.1	1	695	0.49	0.09	0.82
**POL5**	P5 + C4	Covalent	14.2	7.3 ± 1.1	−17	685	0.25	0.06	0.76
**POL6**	P5 + C4 + azide-NH_2_	Covalent	14.2	11.0 ± 0.9	−7	685	0.25	0.06	0.76

*^a^* The mean hydrodynamic diameter (*D*_H_) and zeta potential (*ZP*) were obtained by DLS with intensity of scattered light detected at angle θ = 173° in deionized water (pH ~ 6). *^b^ λ_L_* is the maximum of luminescence emission bands; *Φ_L_*(Ar) and *Φ_L_*(air) are the luminescence quantum yields in argon- and air-saturated dispersions, respectively (excitation wavelength was 400 nm); *F_T_(air)* is the fraction of the triplet states quenched by oxygen in air saturated solutions: *F_T_(air) =* 1 − *Φ_L_(air)/Φ_L_(Ar).*

**Table 4 nanomaterials-12-03350-t004:** Physico-chemical and photophysical characteristics of fresh solutions of polymer-cluster constructs in PBS and their stability after 5 days.

Polymer-Cluster Constructs	*D*_H_ (nm) *^a^*	*ZP* (mV) *^a^*	*λ**_L_* (nm) *^b^*	*Φ_L_*(Ar)	*Φ_L_*(air)	*F_T_(air)*
**POL5, fresh**	7.3 ± 1.1	−17	688	0.27	0.06	0.78
**POL5, 5 days old**	7.9 ± 1.4	−15	690	0.25	0.06	0.76
**POL6, fresh**	11.0 ± 0.9	−7	689	0.27	0.06	0.78
**POL6, 5 days old**	14.2 ± 0.1	−1	690	0.27	0.06	0.78

*^a^* The mean hydrodynamic diameter (*D*_H_) and zeta potential (*ZP*) were obtained by DLS with intensity of scattered light detected at angle θ = 173° in PBS. *^b^ λ_L_* is the maximum of luminescence emission bands; *Φ_L_*(Ar) and *Φ_L_*(air) are the luminescence quantum yields in argon- and air-saturated dispersions, respectively (excitation wavelength was 400 nm); *F_T_(air)* is the fraction of the triplet states quenched by oxygen in air saturated solutions: *F_T_(air)* =1 *− Φ_L_(air)/Φ_L_(Ar).*

## Data Availability

Not applicable.

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
