# Peer review of "Octahedral Molybdenum Cluster-Based Nanomaterials for Potential Photodynamic Therapy"

_nanomaterials, 2022, doi:10.3390/nano12193350_

Round 1
Reviewer 1 Report
The manuscript entitled: "Octahedral molybdenum cluster-based nanomaterials for po-2 tential photodynamic therapy," presents a series of studies the investigated the ability to bond octahedral Molybedenum 6 clusters to various polymers with an aim to improve the lifetime of these complexes within living systems to be used as agents to create singlet oxygen via photochemistry. I think that the manuscript is in general well written and presents some important results. However, with an aim of informing a broader audience, I think that the authors should consider the following improvements and questions before consideration of publication is made.
1. Please formally define the following acronyms: PBS (abstract) and TT.
2. In Fig. 1, I presume that the Mo6 symbol is much smaller relative to the polymers shown in reality. Please note this (e.g., "Mo clusters not to scale").
3. For nonspecialists, what is the source of oxygen that is converted into singlet oxygen? Is it dissolved in the blood? Is it in the breakdown of polymers containing oxygen (e.g. with COOH groups)? No discussion is given of this. In section 2.6.6, there is a brief mention of aqeuous solutions dissolved with air or argon. Air is of course some 80 percent nitrogen. Does nitrogen dissolved in solution affect the results in any way?
4. Why was cholesterol used in this study. Is it to mimic lipids in cell walls? A few sentences explaining this would be very helpful.
5. Define "EPR effect."
6. What is the role of Ar- dispersions? Examination of van der Waals interactions with the polymers?
In summary, I feel that there are some promising results here but the authors need to better explain the motivations to inform a wider audience.
Author Response
The manuscript entitled: "Octahedral molybdenum cluster-based nanomaterials for potential photodynamic therapy," presents a series of studies the investigated the ability to bond octahedral Molybedenum 6 clusters to various polymers with an aim to improve the lifetime of these complexes within living systems to be used as agents to create singlet oxygen via photochemistry. I think that the manuscript is in general well written and presents some important results. However, with an aim of informing a broader audience, I think that the authors should consider the following improvements and questions before consideration of publication is made.
- Please formally define the following acronyms: PBS (abstract) and TT.
Answer: We thank the reviewer for this suggestion. We included the definition of PBS on page 1 and TT on page 4.
- In Fig. 1, I presume that the Mo6 symbol is much smaller relative to the polymers shown in reality. Please note this (e.g., "Mo clusters not to scale").
Answer: We thank the reviewer for this comment. We included this information in the Figure 1 caption (page 3).
- For nonspecialists, what is the source of oxygen that is converted into singlet oxygen? Is it dissolved in the blood? Is it in the breakdown of polymers containing oxygen (e.g. with COOH groups)? No discussion is given of this. In section 2.6.6, there is a brief mention of aqeuous solutions dissolved with air or argon. Air is of course some 80 percent nitrogen. Does nitrogen dissolved in solution affect the results in any way?
Answer: We thank the reviewer for this comment. The singlet oxygen is produced by energy transfer from the triplet states of the light-activated cluster complexes to molecular oxygen dissolved in the aqueous medium (i.e. the intracellular medium). We altered the introduction (page 2) in order to clarify this mechanism.
- Why was cholesterol used in this study. Is it to mimic lipids in cell walls? A few sentences explaining this would be very helpful.
Answer: We appreciate this question. We improved the text in the discussion section (page 7) for better clarity concerning our motivation to use the cholesterol moiety.
- Define "EPR effect."
Answer: We thank the reviewer for this important comment. We included the basic principle of EPR effect in the introduction (page 2).
- What is the role of Ar- dispersions? Examination of van der Waals interactions with the polymers?
Answer: “Ar- dispersions“ stands for argon-saturated aqueous dispersions of the nanoparticles. As singlet oxygen production and luminescence of the cluster complexes are competing processes, the luminescence intensity of Ar saturated (i.e. oxygen-free) dispersions are often compared to that of air saturated dispersions. This allows to characterize the extent of quenching of the triplet states by oxygen and provides an estimation of the efficiency of singlet oxygen production in air saturated dispersions, as the quantum yield of singlet oxygen production in the air should be comparable to (1-Iair/IAr).
- In summary, I feel that there are some promising results here but the authors need to better explain the motivations to inform a wider audience.
We appreciate your helpful comments and we agree with the reviewer that the motivation of this study should be more clear. Therefore, we have added one sentence highlighting that our main goal is to use these systems for tumor-targeted photodynamic therapy (Introduction section, page 2).
Reviewer 2 Report
The authors may check the different meaning of Abstract vs Introduction. The Abstract just describes results - while the introduction is supposed to give all relevant definitions etc This is not the case here! Cf, attached .pdf

Author Response
The authors may check the different meaning of Abstract vs Introduction. The Abstract just describes results - while the introduction is supposed to give all relevant definitions etc This is not the case here! Cf, attached .pdf
- This is the abstract and not the introduction. Don't mix it. Please define "complex of octahedral molybdenum cluster"! please - take some time to think to apply the scientific terminology correctly.
Answer: We thank the reviewer for this important comment. In the abstract (page 1), we removed the information that was more relevant for the introduction section. Also, we improved the text for better clarity concerning the concept of Mo6 clusters, but their constitution – an octahedron of molybdenum atoms surrounded by eight strongly bonded iodine inner ligands and six labile inorganic/organic apical ligands – is explained in the introduction section (page 2).
- Why do you talk about clusters? Are there different ones available? If so, which? .... But - this is part of the introduction!
Answer: We thank the reviewer for this question. We added information in order to provide a better definition of octahedral molybdenum cluster compounds in the introduction section (page 2). Also, their schematic representation is shown in the Figure 2 (B). For stabilization, the clusters are surrounded by eight strongly bonded inner ligands (Li), generally halogen (Cl, Br, I), and six apical ligands (La) that can be whether of organic or inorganic nature to form the cluster complex denoted [Mo6Li8La6]n.
Example: Schematic representation of the molecular structure of the [Mo6Li8La6]n octahedral cluster complex (blue: Mo, W, or Re; magenta: inner ligand Li; green: apical ligand La)
- Concerning Figure 1:
Left structures: Its hydrogen bonding!
Answer: The forces holding the polymer-cluster constructs together are based entirely on electrostatic interactions between the positively charged Mo6 cluster and the negatively charged polymer (top left) or between the negatively charged Mo6 cluster and the positively charged polymer (bottom left).
Middle structures: Do you mean pi-type interaction?
Answer: The structures based on the hydrophobic interactions are probably kept together mostly via van der Waals forces between the cholesterol groups on the polymer and the Mo6 cluster ligands. Pi-type interactions probably do not participate in this case due to the absence of aromatic structures.
Right structure: If you were to talk about complexes of clusters you are talking about dative/coordinative bonds and not covalent bonds.
Answer: The structures on the right really contain covalent bonds formed between the DBCO groups of the polymer and the azide groups of the Mo6 cluster ligands. These covalent bonds originated the strain-promoted azide-alkyne cycloaddition (copper-free click chemistry), resulting in a triazole ring formation.
- Topic 2.5, line 129 from the dpf: “This is nice - but you should place this part into the introduction! “
Answer: We thank the reviewer for this suggestion. We included this information in the last paragraph from the introduction section on page 2 and improved the respective paragraph for better clarity.

Reviewer 3 Report
The manuscript presented for review describes interesing reseach results concerning photodynamic therapy. This work may be published with minor corrections. My comment concerns the cited literature. Authors should add new publications to the references, as no work from 2022 was cited.
Author Response
The manuscript presented for review describes interesting research results concerning photodynamic therapy. This work may be published with minor corrections. My comment concerns the cited literature. Authors should add new publications to the references, as no work from 2022 was cited.
Answer: We thank the reviewer for this suggestion. We have included the following recent publications related to PDT application of Mo6 clusters to better illustrate the current knowledge on this topic.
- B. Smith, L. C. Days, D. R. Alajroush, K. Faye, Y. Khodour, S. J. Beebe, A. A. Holder, Photodynamic Therapy of Inorganic Complexes for the Treatment of Cancer. Photochemistry and Photobiology, 98 (1) (2022) 17. https://doi.org/10.1111/php.13467
- Kirakci, M. Kubáňová, T. PÅ™ibyl, M. Rumlová, J. Zelenka, T. Ruml, K. Lang, A Cell Membrane Targeting Molybdenum-Iodine Nanocluster: Rational Ligand Design toward Enhanced Photodynamic Activity. Inorg. Chem., 61 (12) (2022) 5076. https://doi.org/10.1021/acs.inorgchem.2c00040
- Koncošová, M. Rumlová, R. Mikyšková, M. Reiniš, J. Zelenka, T. Ruml, K. Kirakci, K. Lang, Avenue to X-ray-induced photodynamic therapy of prostatic carcinoma with octahedral molybdenum cluster nanoparticles. J. Mater. Chem. B, 10 (17) (2022) 3303. https://doi.org/10.1039/D2TB00141A